# On the Influence of SGD Hyperparameters on Robustness to Spurious Correlations

## Abstract

Theoretical progress has recently been made in understanding how machine learning models develop reliance on spurious correlations. While empirical findings highlight the influence of stochastic gradient descent (SGD) and its optimization hyperparameters on this behavior, a grounded theoretical explanation remains lacking. Existing theories provide limited justification and fail to account for these phenomena. In this work, we revisit the four-points framework, a widely used theoretical tool for analyzing spurious correlations, to investigate how batch size affects the learning speeds of invariant features in the presence of spurious correlations. Our results show that the framework can account for the faster acquisition of invariant features under small-batch regimes, offering a principled perspective on the role of SGD and its hyperparameters in shaping reliance on spurious correlations. This analysis contributes to a deeper theoretical understanding of the mechanisms underlying robustness and generalization in machine learning.

## 1 Introduction

Modern machine learning models, particularly deep neural networks, have achieved remarkable success across diverse tasks. Yet, this success often masks a critical vulnerability: a tendency to rely on *spurious correlations*. A spurious correlation arises when a feature appears highly predictive of the target in training data but lacks any causal connection. Since such features often provide shortcut decision rules, models are prone to exploiting them rather than relying on the more complex invariant features that ensure robust generalization [Kirichenko et al., 2023].

A well-known example is an image classifier trained to distinguish cows from camels [Beery et al., 2018]. If cows in the training set typically appear in grassy fields and camels in sandy deserts, the model may learn to associate "grass" with "cow" and "desert" with "camel". While effective on the training distribution, this strategy fails when presented with a cow in the desert: the classifier has learned to rely on background texture (spurious feature) rather than the animal's shape (invariant feature). Such failures to generalize out of distribution (OOD) present a major challenge for deploying machine learning systems in safety-critical applications [Ye et al., 2022, 2024].

To better understand this phenomenon, prior works have introduced theoretical models that distill the learning problem into its essential structure. A popular abstraction is the *four-points data model* [Nagarajan et al., 2021], where data lie in a two-dimensional space: one axis corresponds to the invariant feature (e.g., animal type) and the other to the spurious feature (e.g., background). Ideally, a classifier would recover the invariant axis as the decision boundary. However, theoretical analyses show that even in this simplified setting, models systematically exploit spurious correlations. In particular, Nagarajan et al. [2021] identify two main failure modes: (i) *geometric skew*, where max-margin solutions themselves are biased toward spurious features, and (ii) *statistical skew*, where gradient-based optimization amplifies reliance on spurious features even in the absence of geometric

bias. Puli et al. [2023] further study the role of max-margin classifiers in this behavior and also employ the four-point abstraction to model invariant versus spurious dimensions.

Recent research has shifted attention toward the role of stochastic gradient descent (SGD) and its hyperparameters in shaping reliance on spurious features. Gulrajani and Lopez-Paz [2021] show that with appropriate hyperparameter choices, ERM can even surpass more sophisticated domain generalization methods. In particular, training hyperparameters such as learning rate and batch size play a decisive role, as demonstrated by Idrissi et al. [2022], who find that careful tuning can lead to substantial gains in worst-group performance. Barsbey et al. [2025] show that large learning rates can improve robustness to spurious correlations and enhance compressibility of deep models. They attribute this effect to *norm growth* [Merrill et al., 2021], which amplifies updates on minority samples once spurious features are learned [Qiu et al., 2024, Yang et al., 2024]. Similarly, Mirzaie et al. [2025] argue that implicit regularization induced by SGD [Smith et al., 2021] makes small batch sizes and large learning rates more robust to spurious correlations. Together, these studies suggest that optimization hyperparameters strongly influence the balance between invariant and spurious feature learning.

Nevertheless, theoretical understanding of this link remains limited. Existing explanations either assume infinitesimal learning rates, as in Nagarajan et al. [2021], Ye et al. [2023], or rely on norm growth arguments without providing a principled connection between SGD hyperparameters and reliance on spurious correlations, as in Barsbey et al. [2025]. This gap motivates the search for frameworks that can capture the concrete effects of learning rate and batch size in practice.

**Our contribution.** In this work, we aim to bridge this gap by revisiting the *four-points data model*, a minimal setting for studying spurious correlations. We use this framework to conduct a fine-grained analysis of SGD dynamics. Specifically, we investigate how the choice of batch size influences the learning speeds of invariant features. Our main finding is that smaller batch sizes can, under certain conditions, promote faster learning of the invariant feature, providing a theoretical justification for this phenomena. This contributes to a more principled understanding of how optimization choices shape model generalization and robustness. Our results highlight the potential of the four-points framework as a foundation for deeper theoretical analysis of SGD hyperparameters, and its extension to richer models with additional dimensions for memorization [Puli et al., 2023, Qiu et al., 2024].

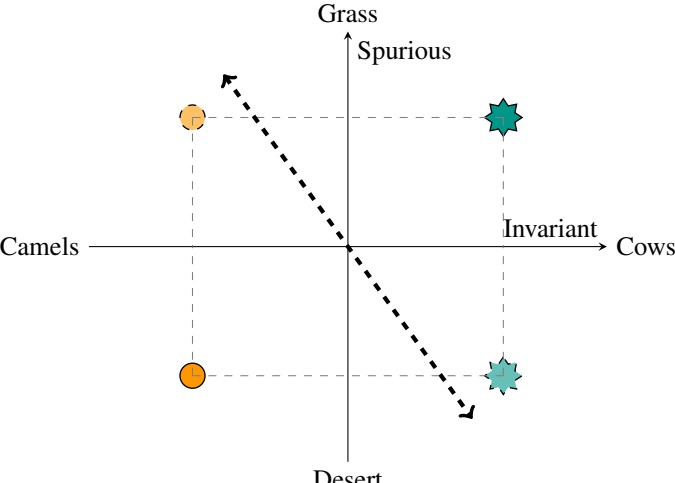

Figure 1: Conceptual illustration of spurious correlation. An ideal classifier (e.g., max-margin) should separate data based on the invariant feature (a vertical line separating Cows vs. Camels). However, models trained with standard methods often learn a decision boundary that relies on the spurious feature (e.g., a diagonal line separating Grass vs. Desert), leading to failures on out-of-distribution examples (a cow in the desert or a camel on grass).

## 2 Four-Points Data Model

We study the standard four-points model used to analyze spurious correlations [Nagarajan et al., 2021]. Each data point lies in $\mathbb{R}^2$ and is written as $(x_i, x_s)$, where

$$x_i \in \{-1, +1\}, \qquad x_s \in \{-B, +B\}, \qquad B > 1.$$

The label is defined by $y = x_i$, so the first coordinate is the *invariant feature* and the second coordinate is the *spurious feature*. The data-generating process is

$$x_i \sim \text{Rad}, \tag{1}$$

$$x_s \mid x_i = \begin{cases} +Bx_i & \text{with probability } p, \\ -Bx_i & \text{with probability } 1-p, \end{cases} \tag{2}$$

where $\text{Rad}$ denotes a Rademacher random variable (i.e. $\Pr(x_i = +1) = \Pr(x_i = -1) = 1/2$), and $p \in [1/2, 1]$ measures the strength of the spurious correlation. Thus the support of the distribution is the four points $\{-1, +1\} \times \{-B, +B\}$ and, for each class $x_i$, a proportion $p$ of class samples have the spurious feature aligned with the invariant feature while proportion $1 - p$ are anti-aligned. The four-points model is minimal yet expressive enough to capture the qualitative mechanisms by which models can rely on spurious features.

## 3 Setup

We consider a linear classifier parameterized by weights $(w_i, w_s) \in \mathbb{R}^2$:

$$f_w(x_i, x_s) = w_i x_i + w_s x_s. \tag{3}$$

We use the hinge loss

$$\ell\big(f_w(x), y\big) = \max\{0, \, 1 - y f_w(x)\}. \tag{4}$$

Because the hinge loss is piecewise linear, the gradient (subgradient) with respect to the weight vector is simple. For a single example $(x_i, x_s, y)$ define the margin

$$m := y f_w(x) = y(w_i x_i + w_s x_s).$$

If $m < 1$ (the example is *active*), the (sub)gradient is

$$\nabla_{w_i} \ell = -y x_i, \qquad \nabla_{w_s} \ell = -y x_s, \tag{5}$$

and if $m \geq 1$ then the gradients are zero. Consequently, a single-example (batch size 1) SGD update with learning rate $\eta > 0$ is

$$w \leftarrow w - \eta \nabla_w \ell = \begin{cases} w + \eta \, y \begin{pmatrix} x_i \\ x_s \end{pmatrix}, & \text{if } y f_w(x) < 1, \\ w, & \text{if } y f_w(x) \geq 1. \end{cases} \tag{6}$$

For a full-batch gradient step we instead average the contributions of all active points.

**Definition 3.1** (Regular SGD run)**.** We call a single-example SGD run *regular* if minority examples do not appear consecutively more than once — i.e. every minority example is followed by at least one majority example before the next minority appears.

Regular SGD run has high probability when $p$ is close to 1, and the phrase is only used to simplify discussion of typical sample-ordering effects under SGD.

## 4 Role of Batch Size in the Speed of Learning the Invariant Feature

Intuitively, batch size affects the *ordering* and *averaging* of gradient contributions from aligned (majority) and anti-aligned (minority) points; because the hinge loss is thresholded at margin 1, this ordering can change which points remain active and thus the subsequent updates. We formalize this observation for the early phase of training.

**Proposition 1.** *Let $(w_i, w_s)$ be initialized at $(0, 0)$. Suppose the learning rate $\eta$ satisfies*

$$\eta > \frac{1}{1 + (2p - 1)B^2}.$$

*Then, until minority points become inactive, the growth rate of the invariant weight $w_i$ is larger under single-sample SGD with regular optimization than under full-batch gradient descent.*

*Proof.* The update applied to the invariant feature at step $t$ depends on whether the majority and minority samples incur positive loss. We compute the updates for each setting.

**Full-batch.** At initialization, when neither minority nor majority samples have zero loss, the updates on $w_i$ and $w_s$ are

$$w_i^{t+1} = w_i^t + \eta y x_i = w_i^t + \eta, \tag{7}$$

$$w_s^{t+1} = w_s^t + \eta\big(pB - (1 - p)B\big) = w_s^t + \eta(2p - 1)B. \tag{8}$$

In this case, for majority and minority samples we have

$$y(f_{w^{t+1}}(x_{\mathrm{maj}}) - f_{w^t}(x_{\mathrm{maj}})) = \eta\big(1 + (2p - 1)B^2\big) > 1. \tag{9}$$

By initializing at 0, the loss for majority samples becomes zero after the first iteration.

Now let us analyze the case where the loss of majority samples is zero, but the minority loss is not. The updates are

$$w_i^{t+1} = w_i^t + \eta(1 - p), \qquad w_s^{t+1} = w_s^t - \eta(1 - p)B. \tag{10}$$

In this case, the change in margins is

$$y(f_{w^{t+1}}(x_{\mathrm{maj}}) - f_{w^t}(x_{\mathrm{maj}})) = -\eta(1 - p)(B^2 - 1) < 0, \tag{11}$$

$$y(f_{w^{t+1}}(x_{\mathrm{min}}) - f_{w^t}(x_{\mathrm{min}})) = \eta(1 - p)(B^2 + 1) > 0. \tag{12}$$

If the loss value for minority samples does not vanish, this case continues for

$$\left\lceil \frac{\eta(1 + (2p - 1)B^2) - 1}{\eta(1 - p)(B^2 - 1)} \right\rceil$$

steps. At each step, $w_i$ increases by $(1 - p)\eta$. Once $f(x_{\mathrm{maj}})$ falls below 1, the update rule from Equation 7 applies again.

To summarize, until the loss of minority samples becomes zero, the effective update rate of $w_i$ lies in the range

$$\frac{\frac{\eta(1 + (2p-1)B^2) - 1}{(B^2 - 1)} + \eta}{\frac{1 + (2p-1)B^2}{(1-p)(B^2-1)} + 1} \leq \dot{w}_i \leq \frac{\frac{\eta(1 + (2p-1)B^2)}{(B^2-1)} + \eta}{\frac{\eta(1 + (2p-1)B^2) - 1}{\eta(1-p)(B^2-1)} + 1}. \tag{13}$$

Therefore, we have the approximation

$$\dot{w}_i^{FB} \approx (1 - p)\eta. \tag{14}$$

**Minibatch of size 1.** We now analyze the other extreme case. When the batch size equals 1, with probability $1 - p$ the sample is a minority point, yielding the updates

$$w_i^{t+1} = w_i^t + \eta, \qquad w_s^{t+1} = w_s^t - \eta B. \tag{15}$$

The same update for $w_i$ also holds when a majority sample with positive loss is drawn. Starting from zero initialization, we obtain $w_i = n\eta$ for some $n \in \mathbb{N}^{\geq 0}$. Similarly, in the regular optimization setting, we can write $w_s = Bm\eta$ with $m \in \mathbb{N}^{\geq -1}$.

Since $\eta > \frac{1}{1 + (2p-1)B^2} \geq \frac{1}{1+B^2}$ and $w_i^{t=0} = w_s^{t=0} = 0$, if at some point $m \geq 1$, then

$$y(f(x_{\mathrm{maj}})) = \eta(n + B^2 m) > 1, \tag{16}$$

which implies that $m$ remains confined to $\{-1, 0, 1\}$.

During training, after each update on a minority sample, the next step is a majority sample (by regular optimization). In both updates, $w_i$ increases by $\eta$. Since minority samples occur with frequency $(1 - p)$, the effective update rate of $w_i$ is at least

$$\dot{w}_i^{MB} = 2(1 - p)\eta. \tag{17}$$

126 **Comparison.** Comparing the two cases, we conclude that

$$\dot{w_i}^{MB} \geq 2(1-p)\eta \quad > \quad (1-p)\eta \approx \dot{w_i}^{FB}. \tag{18}$$

127 Hence, the invariant feature is learned faster with minibatch size 1 (under regular optimization) than
128 with full-batch training. □

## 5 Discussion

130 Proposition 1 shows that when the learning rate is sufficiently large, the invariant feature is learned
131 at a faster rate. This insight helps explain why smaller batch sizes often lead to greater robustness
132 against spurious correlations. Prior work has established a close connection between reliance on
133 spurious features and the relative speed of learning the invariant feature [Yang et al., 2024, Qiu et al.,
134 2024, Joshi et al., 2023].

135 The lower-bound condition on the learning rate is also consistent with earlier findings that larger
136 learning rates improve robustness to spurious correlations. It is worth noting that the implicit
137 regularization properties of SGD depend not only on the learning rate but also on its ratio to batch
138 size. Moreover, when the learning rate approaches zero, the difference between full-batch and
139 minibatch gradient descent effectively disappears, since all updates converge to the same trajectory.
140 This limitation underlies the framework of Nagarajan et al. [2021], Ye et al. [2023], which assumes
141 infinitesimal step sizes.

142 The choice of loss function also plays an important role. In our analysis, hinge loss was particularly
143 useful because of its bounded derivative, which made it possible to precisely track updates. Other loss
144 functions, such as cross-entropy with sigmoid activation, also share this bounded-derivative property,
145 suggesting that similar results should extend beyond hinge loss.

146 Since the deactivation of majority samples is a primary factor underlying the slower learning of
147 the invariant feature, we conjecture that when the loss function exhibits sharper behavior—such as
148 when using a softmax with a high temperature—the influence of smaller batch sizes on robustness to
149 spurious correlations becomes more pronounced. Although a formal proof is left for future work, we
150 further conjecture that, under certain conditions, training with single-sample batches converges to
151 $w_s = 0$, whereas this need not hold in the full-batch regime.

152 Finally, to more closely reflect real-world scenarios and complex models such as deep neural
153 networks—which have the capacity to *memorize* individual samples without learning the underlying
154 structure—future extensions of our framework should incorporate additional noise dimensions [Puli
155 et al., 2023, Qiu et al., 2024]. This would allow a more complete characterization of how SGD
156 hyperparameters and model complexity interact in shaping reliance on spurious correlations.

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
