# OpenReview forum: "On the Influence of SGD Hyperparameters on Robustness to Spurious Correlations"
_EurIPS.cc/2025/Workshop/UPLB — UPLB2025_

### Official Review · Reviewer_yyLi · 2025-10-25
**On the Influence of SGD Hyperparameters on Robustness to Spurious Correlations**

**Rating:** 5
**Confidence:** 4

**Review:**

In this manuscript the authors investigate the role of the batch size of stochastic gradient descent on a linear classification problem in dimension 2 where data have a particular structure of correlation (presence of spurious correlations that may substantially affect of the learned classifier). The work is within the scope of the workshop where it has been submitted. The hypersimplified setting allows the exact analysis of SGD and to conclude that large SGD noise (small batch) is beneficial for the performances when spurious correlations are present. While the conclusion is relevant, the setting is very much not realistic in the sense that the model is (i) a linear classifier, (ii) trained on a specific loss (hinge loss), (iii) with a specific data distribution and in very low dimension (2 dimensional input data). I think that among the assumptions, low dimensionality is really oversimplifying the problem. I think that an analysis, at least numerical (simulations), on a higher dimensional settings, showing the same trend would have improved the manuscript.

---

### Decision · Program_Chairs · 2025-11-03

Accept (Poster)